# Diet-Induced Hypercholesterolemia Leads to Cardiac Dysfunction and Alterations in the Myocardial Proteome

**DOI:** 10.3390/ijms23137387

**Published:** 2022-07-02

**Authors:** Márton Richárd Szabó, Márton Pipicz, Márta Sárközy, Bella Bruszel, Zoltán Szabó, Tamás Csont

**Affiliations:** 1Department of Biochemistry, Albert Szent-Györgyi Medical School, University of Szeged, H-6720 Szeged, Hungary; szabo.marton@med.u-szeged.hu (M.R.S.); pipicz.marton@med.u-szeged.hu (M.P.); sarkozy.marta@med.u-szeged.hu (M.S.); 2Interdisciplinary Centre of Excellence, University of Szeged, H-6720 Szeged, Hungary; bruszelbella@gmail.com (B.B.); szabo.zoltan@med.u-szeged.hu (Z.S.); 3Institute of Medical Chemistry, Albert Szent-Györgyi Medical School, University of Szeged, H-6720 Szeged, Hungary

**Keywords:** hypercholesterolemia, myocardial proteomics, network analysis, cardiac dysfunction, mitochondrial respiratory chain, contractile proteins

## Abstract

Elevated blood cholesterol is a major risk factor for coronary heart disease. Moreover, direct effects on the myocardium also contribute to the adverse effects of hypercholesterolemia. Here, we investigated the effect of hypercholesterolemia on the cardiac proteome. Male Wistar rats were fed with a laboratory rodent chow supplemented with 2% cholesterol for 8 weeks to induce hypercholesterolemia. The protein expression data obtained from the proteomic characterization of left ventricular samples from normo- and hypercholesterolemic animals were subjected to gene ontology (GO) and protein interaction analyses. Elevated circulating cholesterol levels were accompanied by diastolic dysfunction in cholesterol-fed rats. The proteomic characterization of left ventricular samples revealed altered expression of 45 proteins due to hypercholesterolemia. Based on the Gene Ontology analysis, hypercholesterolemia was associated with disturbed expression of cytoskeletal and contractile proteins. Beta-actin was downregulated in the hypercholesterolemic myocardium, and established a prominent hub of the protein interaction network. Analysis of the unfiltered dataset revealed concordant downregulated expression patterns in proteins associated with the arrangement of the contractile system (e.g., cardiac-specific troponins and myosin complex), and in subunits of the mitochondrial respiratory chain. We conclude that the observed changes in the cardiac proteome may contribute to the development of diastolic dysfunction in hypercholesterolemia.

## 1. Introduction

Metabolic disorders, e.g., dyslipidemia, diabetes mellitus, or metabolic syndrome, are major risk factors for ischemic heart diseases [1]. Imbalance in the lipid metabolism, especially sustained hypercholesterolemia, contributes to the development of cardiovascular diseases through the formation of atherosclerosis and subsequent acute ischemic events [2]. Moreover, independent of the vascular effects, hypercholesterolemia exerts direct adverse effects on the myocardium, which further impair cardiac function and stress tolerance [3,4,5,6]. Previous studies reported the presence of mild diastolic dysfunction and blunted adaptation to ischemia in rodent models of diet-induced hypercholesterolemia [7,8].

Despite the well-known adverse cardiac effects of hypercholesterolemia, the underlying molecular mechanisms and involved pathways are still not fully understood. Previous studies suggested that deteriorated mitochondrial function was possibly related to the adverse cardiac effects of hypercholesterolemia [9,10]. Moreover, endothelial dysfunction and enhanced oxidative damage-related mechanisms, such as increased NADPH oxidase activity [3,4], were also implied in the direct cardiac consequences of hypercholesterolemia.

The recent advances in the field of ‘omics’ methods provide a great and effective tool for the high-throughput screening and molecular profiling of different diseases. Microarray screening revealed transcriptomic alterations in the myocardium induced by hyperlipidemia and metabolic syndrome, showing altered expression of genes related to energy production, structural development, and stress [11,12,13]. However, to date, there are no comprehensive data regarding the global left ventricular proteome changes in the setting of hypercholesterolemia. Therefore, in the present study, we aimed to investigate the global changes in the left ventricular proteome in a rat model of diet-induced hypercholesterolemia using shotgun proteomic analyses. Subsequently, bioinformatic analyses (pathway enrichment and protein interaction analyses) were performed on the proteomic data in order to elucidate the potential underlying molecular mechanisms and pathways regarding the direct adverse cardiac effects of hypercholesterolemia.

## 2. Results

### 2.1. Eight Weeks of Cholesterol-Enriched Diet in Rats Resulted in Elevated Plasma Lipid Levels

At the end of an 8-week feeding period, there was no difference in the body weight and the left ventricular tissue weight of the animals receiving a cholesterol-enriched diet or a standard diet (Table 1). However, the total plasma cholesterol was markedly elevated in the cholesterol-fed group, supporting the manifestation of diet-induced hypercholesterolemia. Interestingly, the plasma total triacylglycerol level also increased significantly in the hypercholesterolemic group compared with the control rats.

### 2.2. Cholesterol-Enriched-Diet-Induced Diastolic Dysfunction in the Heart

Transthoracic echocardiography was performed at the end of the feeding period to investigate the morphological and functional effects of diet-induced hypercholesterolemia on the myocardium. After 8 weeks of the diet, the cardiac morphology remained unaffected, as shown by the systolic and diastolic wall thickness parameters (Table 2). Furthermore, there were no differences in the left ventricular end-diastolic and end-systolic diameters, fractional shortening, and ejection fraction, respectively, which indicate preserved left ventricular systolic function (Table 2). Interestingly, the heart rate was significantly decreased in the settings of hypercholesterolemia. The early (E) and late (A) ventricular filling velocities showed a trend toward a decrease; however, the E/A ratio was significantly elevated in the hearts of hypercholesterolemic animals, suggesting impaired diastolic function (Table 2). The presence of diastolic dysfunction was further supported by the significantly decreased values of the mitral annulus velocity (e’) and mitral valve deceleration time in the hypercholesterolemic rats. Nevertheless, the E/e’ ratio was not affected significantly (Table 2).

### 2.3. General Proteomic Characterization of the Left Ventricle of Hypercholesterolemic Rats

Altogether, 901 proteins were reliably identified from left ventricular samples by means of mass spectrometry. Statistical analysis (Welch’s *t*-test) performed on the identified proteins revealed altered levels (*p* < 0.05) of 75 proteins due to hypercholesterolemia. Proteins showing *p* < 0.05 and >1.2 or <0.83-fold changes in response to hypercholesterolemia were considered as significant alterations and used for further network analysis. Based on these criteria, we observed the upregulation of 23 proteins and downregulation of 22 in the left ventricle of hypercholesterolemic animals compared with the normocholesterolemic controls (Figure 1; Table 3).

### 2.4. Pathway Enrichment Analysis of the Significantly Altered Proteins Revealed Changes in the Contractile and Cytoskeletal Systems

In order to assign biological functions and reveal potential networks for the significantly changed proteins, Gene Ontology (GO) and subsequent pathway enrichment analyses were carried out. The GO analysis covered all three independent ontology categories, including molecular function (MF), biological process (BP), and cellular component (CC). We observed the enrichment of 31 GO terms at FDR < 0.1, altogether (Figure 2). Interestingly, according to the GO terminology, a substantial number of the enriched nodes were in the CC category, possibly indicating hypercholesterolemia-induced rearrangements of subcellular structures and macromolecular complexes in the left ventricle. Based on the GO analysis, the enriched ontology terms involved proteins associated with contractile function and cytoskeletal organization (Figure 2). Additionally, a minor enrichment of mitochondrial proteins was also observed (Figure 2). Overall, the involvement of the observed processes might contribute to the initiation of the subcellular structural remodeling and subsequent contractile impairment of the myocardium in hypercholesterolemic animals.

### 2.5. Functional Interaction Analysis of the Differentially Expressed Proteins

As the next step, we assessed the functional interactions and conducted subsequent cluster analysis within the significantly altered proteins. Our analyses revealed a modest number of connections among the identified proteins (Figure 3). Interestingly, beta-actin (ACTB) was downregulated in the hypercholesterolemic myocardium and ACTB established a prominent hub of the revealed network (Figure 3). ACTB turned out to be connected to both structural and accessory proteins. Among these interactions, many proteins were upregulated, such as Actin-related protein 2/3 complex subunit 1A (ARPC1A), Adenylyl cyclase-associated protein 1 (CAP1), WD repeat-containing protein 1 (WDR1), Myosin-7 (MYH7), Gelsolin (GSN), and Ras homolog family member A (RHOA). At the same time, the levels of Myosin-6 (MYH6), Myosin heavy chain 14 (MYH14), Collagen type 1 alpha 2 chain (COL1A2), and Cytoplasmic dynein heavy chain 1 (DYNC1H1) were downregulated (Figure 3).

Additionally, the cluster analysis also revealed other minor subnetworks among the resulting interactions (Figure 3), which might implicate disturbed metabolic functions and subsequent energy production. For instance, the resulting functional interaction network contained metabolic enzymes with altered expressions, including Pyruvate carboxylase (PC), Phosphoglycerate mutase 1 (PGAM1), Transketolase (TKT), Creatine kinase (CKB), Malic enzyme 1 (ME1), and Lactate dehydrogenase (Figure 3). Moreover, three subunits of the mitochondrial NADH dehydrogenase complex (NDUFV3, NDUFA3, and LOC684509) showed significantly downregulated expression and formed one of the loops in the network (Figure 3). These subnetworks may suggest impaired mitochondrial function and imbalance in the energy production of the hypercholesterolemic myocardium.

### 2.6. Protein-Specific Gene Set Enrichment Analysis Revealed Downregulated Expression Patterns of Mitochondrial and Contractile Proteins in the Unfiltered, Whole Left Ventricular Proteome

To reveal the potential associations between hypercholesterolemia-induced cardiac phenotype and classes of similarly changed proteins in the heart, gene set enrichment analysis (GSEA) was performed on the unfiltered, whole proteomic dataset. The GSEA identified similar downregulated expression patterns among the proteins related to specific pathways. The most prominent enrichment was observed in the GO terms associated with mitochondrial complexes, with particular emphasis on the elements of the respiratory chain complexes (Figure 4A). Additionally, our analysis revealed downregulated expression patterns of proteins previously assigned to heart development processes in the GO terminology (Figure 4A). Then, subsequent leading-edge analysis was carried out to determine which subsets of proteins contributed the most to the enriched GO terms. As expected, proteins of the respiratory chain complexes, i.e., subunits of the NADH:Ubiquinone Oxidoreductase complex (NDFU), were among the leading enrichment set (Figure 4B). Furthermore, another set of leading-edge proteins could be identified with important roles in normal cardiac contractile function (Figure 4B). For instance, downregulated expression patterns were observed in the case of cardiac-specific isoforms of the troponin complex, such as Troponin T2 (TNNT2), Troponin C1 (TNNC1), Troponin I3 (TNNI3), and Tropomyosin 1 (TPM1). Furthermore, hypercholesterolemia seemed to negatively affect the protein expression pattern of the ventricular isoform of myosin light chain (MYL3), as well as myosin heavy chain 6 (MYH6), which is preferably expressed in the ventricles of smaller mammals with rapid heart rates.

### 2.7. KEGG Analysis of the Output of GSEA

In order to categorize the output protein list of the previous GSEA analysis, the core enrichment proteins from the significantly enriched gene sets were further analyzed according to the in-built Kyoto Encyclopedia of Genes and Genomes (KEGG) of the Pathview package. Proteins related to cardiac muscle contraction were affected, as shown by the concordant downregulated expression patterns of cardiac-specific troponins and myosin complex in the left ventricle of hypercholesterolemic animals (Figure 5). Additionally, the results of the KEGG-based analysis of our protein sets further support the possible deterioration of the mitochondrial function in the hypercholesterolemic left ventricle, affecting the expression of components of all the five major complexes of the respiratory chain system responsible for oxidative phosphorylation (Figure 6).

## 3. Discussion

In the present study, we investigated the effect of diet-induced hypercholesterolemia on the left ventricular proteome. To the best of our knowledge, this is the first proteomic study focusing on the myocardium of hypercholesterolemic animals. Our results show that chronic hypercholesterolemia alters the level of cardiac proteins related to the maintenance of cytoskeletal structure and energy-generation processes. Moreover, our enrichment and functional interaction analyses revealed solid networks among the identified proteins, thereby providing new aspects and deeper insights into the potential underlying subcellular mechanisms of the direct myocardial effects of hypercholesterolemia leading to cardiac dysfunction.

The general characterization and echocardiographic parameters demonstrating diastolic dysfunction in our experimental model are in accordance with previous reports from our laboratory and others using rodent models of diet-induced hypercholesterolemia [3,4,10,14,15]. In the present study, eight-week excess cholesterol uptake elevated the circulating cholesterol and resulted in diastolic cardiac dysfunction without any significant morphological changes in the heart. Previous studies suggested that hyperlipidemia, especially hypercholesterolemia, leads to the augmentation of oxidative and nitrosative stress and enhanced proinflammatory cytokine production (e.g., TNF-α and IL-6), consequently contributing to cardiac and endothelial dysfunction [3,4,16]. The related oxidative damage possibly involves contractile and cytoskeletal proteins and, hence, likely contributes to contractile impairment. The direct impact of increased plasma cholesterol on cardiac function is supported by both human and experimental animal investigations [8]. Echocardiographic characterization of patients with primary hypercholesterolemia or familial hypercholesterolemia without a history of cardiovascular disease disclosed subclinical myocardial abnormalities and contractility impairment [17,18]. Likewise, diminished cardiac function was also observed in hypercholesterolemic animals in vivo [19,20]. In accordance with the literature, our measured echocardiographic parameters suggest impaired diastolic function [21], while the ventricular wall thicknesses remained unaffected in the hypercholesterolemic myocardium, similar to previous studies [4,22].

Proteomic characterization of the hypercholesterolemic myocardium in the current study revealed modest quantitative changes in a substantial number of proteins, which is in accordance with other proteomics-based studies of the left ventricle with impaired contractile function [23,24]. One of the most notable changes in the level of an individual protein was observed in the case of the downregulated expression of ACTB, an essential component of the non-contractile cytoskeleton system. Similar to our results, metabolic perturbations, such as hyperglycemia and hyperlipidemia, reduced ACTB in both ventricles, which was associated with the further impairment of cellular elasticity and disorganized myocardial actin cytoskeleton [25,26,27]. Furthermore, in our study, MYH6 decreased, and MYH7 increased in the heart as a result of cholesterol feeding. The MYH6/MYH7 ratio is considered as a descriptive indicator of cardiac function, and the shift from MYH6 to MYH7 resulting in a decreased ratio indicated a maladaptive change in cardiac diseases [28,29,30,31]. The actin-activated ATPase activity of MYH6 is higher than that of MYH7, so both the relative and absolute repression of MYH6 may lead to compromised heart function [32].

Based on our GO analysis of the differentially expressed proteins, the resulting network indicates alterations in the contractile apparatus and cytoskeletal system of the hypercholesterolemic left ventricle. To date, this study is the first to propose that quantitative changes in myofibrillar proteins might also be responsible for the direct cardiac effects of hypercholesterolemia. Additionally, the protein–protein interaction network analysis of the significantly altered proteins further affirmed the influence of high cholesterol levels on cytoskeletal organization and structural development in the left ventricle. Interaction analysis showed that ACTB formed the hub of the revealed network, being involved in many interactions with other accessory proteins (e.g., with upregulated CAP1, WDR1, GSN, RHOA, and ARPC1A), suggesting myocardial cytoskeleton rearrangement in response to hypercholesterolemia. These findings are in accordance with studies indicating the crucial role of actin assembly and disassembly dynamics in heart diseases [33,34].

A deeper analysis of the whole, unfiltered left ventricular proteome demonstrated that many protein sets are downregulated in the hypercholesterolemic heart. In accordance with the GO analysis restricted to the list of significantly altered proteins, contractile proteins showed similarly downregulated expression patterns in the hypercholesterolemic heart. Moreover, our results showed that hypercholesterolemia negatively influenced many protein components of the respiratory chain system in the heart, potentially leading to disturbances in energy supply and consequent contractile impairment. Although the individual expression of the majority of proteins in the enriched protein sets failed to meet the criteria of a significant change, the coordinated downregulation of these proteins as a group was significant. These findings are in line with other reports where impaired mitochondrial function was implicated in metabolic diseases, such as diabetes, obesity, and hypertension, accompanied by heart failure and dysfunction [35,36]. Similarly, the impairment of mitochondrial function is also implicated in the adverse cardiac effects of hypercholesterolemia [37]. Mitochondrial cholesterol accumulation is suggested to be a decisive factor for mitochondrial dysfunction, as increased cholesterol levels interfere with the normal function of mitochondrial membrane proteins and transporters [38]. Additionally, disturbed myocardial ATP synthesis and bioenergetics impairment are documented outcomes of a high-cholesterol diet [10], which might be associated with the downregulation of the elements of the respiratory chain complex.

Similar to all experimental investigations, our study is not without limitations. First of all, since we have applied a rodent model of hypercholesterolemia in our present study, further confirmation of our results is urged in the future in humans due to potential species-dependent differences in the cardiac effects of hypercholesterolemia. Moreover, we applied a shotgun proteomic analysis in the present study, which may have higher limits of detection than targeted approaches; however, it provides the possibility to obtain quantitative information about as many proteins as possible. Our study and the applied protocols were focused on the detection of differences in the expression of proteins, and the identification of possible changes in posttranslational modifications of proteins was outside of the scope of the present study. Future proof-of-concept studies are recommended to further confirm the causal role of the proteins or network of proteins identified in our study in diastolic dysfunction in the heart using targeted proteomics or immunochemical methods.

## 4. Materials and Methods

### 4.1. Animals

Altogether, 12 adult male Wistar rats were used in this study. The animals were kept in pairs in individually ventilated cages in a temperature-controlled room with 12 h:12 h light/dark cycles. Laboratory chow and water were supplied ad libitum throughout the study. The experiment conformed to the Guide for the Care and Use of Laboratory Animals published by the US National Institutes of Health (NIH publication No. 85-23, revised 1996) and was approved by the Animal Research Ethics Committee of the University of Szeged (approval code: XV.1181/2013-2018).

### 4.2. Experimental Setup

Hypercholesterolemia was induced as described previously [10,39]. Male Wistar rats were fed with laboratory chow enriched with 2% (*w/w*) cholesterol (Hungaropharma Zrt., Budapest, Hungary) and 0.25% (*w/w*) sodium–cholate hydrate (Sigma, St. Louis, MO, USA) for 8 weeks (*n* = 6), while rats fed with standard laboratory chow were used as the control (*n* = 6). At the end of the diet period, the rats were anesthetized by the intraperitoneal injection of pentobarbital sodium (Euthasol; 50 mg/kg; Produlab Pharma b.v., Raamsdonksveer, The Netherlands). Blood samples were collected from the thoracic aorta for plasma lipid measurements, and then hearts were immediately excised and placed in ice-cold Krebs–Henseleit buffer. The isolated hearts were cannulated and perfused with oxygenated Krebs–Henseleit buffer at 37 °C, according to Langendorff, in order to eliminate blood from the coronary vessels [39,40]. After 5 min of perfusion, the left ventricular tissue was frozen and stored at −80 °C until proteomic analysis. The body weight and tibia length were also measured.

### 4.3. Plasma Lipid Measurement

Blood samples were collected in EDTA-containing blood collection tubes. The plasma was separated by centrifugation (3000× *g* for 15 min at 4 °C). The upper, cell-free phase was used to determine the total cholesterol and triglyceride concentrations using a colorimetric detection method (Diagnosticum, Budapest, Hungary) with a microplate reader (BMG Labtech, Ortenberg, Germany). The plasma total cholesterol and triglyceride measurements were performed as described previously [39].

### 4.4. Transthoracic Echocardiography

Cardiac morphology and function were assessed by transthoracic echocardiography at week 8, as described previously [22,24]. The rats were anesthetized with 2% isoflurane (Forane, AESICA, Queenborough Limited, Kent, UK). Two-dimensional, M-mode, Doppler, tissue Doppler, and four-chamber-view images were performed according to the criteria of the American Society of Echocardiography with a Vivid IQ ultrasound system (General Electric Medical Systems, New York, NY, USA) using a phased array 5.0–11 MHz transducer (GE 12S-RS probe, General Electric Medical Systems, New York, NY, USA). Data of three consecutive heart cycles were analyzed (EchoPac Dimension software v201, General Electric Medical Systems, New York, NY, USA) by an experienced investigator in a blinded manner. The mean values of the three measurements were calculated and used for statistical evaluation. Systolic and diastolic wall thicknesses were obtained from the parasternal short-axis view at the level of the papillary muscles (anterior and inferior walls) and the long-axis view at the level of the mitral valve (septal and posterior walls). The left ventricular diameters were measured by means of M-mode echocardiography from long-axis and short-axis views between the endocardial borders. Functional parameters, including the ejection fraction and fractional shortening, were calculated on M-mode images in the long-axis view. Diastolic function was assessed using pulse-wave Doppler across the mitral valve and tissue Doppler on the septal mitral annulus from the apical four-chamber view. The early (E) and atrial (A) flow velocities, as well as septal mitral annulus velocity (e’), indicated diastolic function.

### 4.5. Protein Extraction

Approximately 30 mg of powered left ventricular tissue samples were homogenized in lysate buffer (containing 2% Sodium dodecyl sulfate (SDS) and 0.1 M Dithiothreitol (DTT) in 0.1 M Tris solution). The homogenized samples were incubated at 98 °C for 5 min. Proteins were precipitated by the addition of a methanol/chloroform mixture (4:1) and were resuspended in 8 M urea. The total protein contents were determined using the BCA (Thermo Scientific, Waltham, MA, USA) protocol.

### 4.6. Protein Digestion

Homogenates of all individual pulverized tissue samples containing 20 µg protein were diluted to 30 µL with 0.1 M NH_4_HCO_3_ (pH = 8.0) buffer, and then 15 µL 0.1% RapiGest SF (Waters) and 2 µL 100 mM DTT solution were added and the mixture was stored at 60 °C for 30 min to unfold and reduce the proteins. A volume of 2 µL 200 mM iodo-acetamide (IAA) solution was added to alkylate the proteins, which were kept for an additional 30 min in the dark at room temperature. The samples were digested with trypsin (Promega, enzyme/protein ratio: 0.4/1) for 3 h at 37 °C. The digestion was stopped by the addition of 1 µL of concentrated hydrochloric acid. Pooled samples were created by mixing equal amounts of all digested samples to build a spectral library for quantitative liquid chromatography–mass spectrometry (LC-MS) analysis of individual samples.

### 4.7. LC-MS Analysis

The separation of the digested samples was carried out on a nanoAcquity Ultra Performance Liquid Chromatograph (UPLC, Waters, Milford, MA, USA) using a Waters ACQUITY UPLC M-Class Peptide C18 (130 Ǻ, 1.78 µm, 75 µm × 250 mm) column with a 90 min gradient. The eluents were water (A) and acetonitrile (B) containing 0.1 *v/v*% formic acid, and the separation of the peptide mixture was performed at 45 °C with a 0.35 µL/min flow rate using an optimized nonlinear LC gradient (3%–40% B). The LC was coupled to a high-resolution Q Exactive Plus quadrupole-orbitrap hybrid mass spectrometer (Thermo Scientific, Waltham, MA, USA). The measurements of the digested samples were performed in the DIA (Data Independent Acquisition) mode. The survey scan for the DIA method operated with a 35,000 resolution. The full scan was performed between 380 and 1020 m/z. The AGC target was set to 5 × 10^6^ or 120 ms maximum injection time. In the 400–1000 m/z region, 22 m/z-wide overlapping isolation windows were acquired at 17,500 resolution (AGC target: 3 × 10^6^ or 100 ms injection time, normalized collision energy: 30 for charge 2). Gas-phase fractionated (GPF) DIA measurements of the pooled samples were used to build a spectral library using the same settings, except for the m/z range. Four GPF LC-MS analyses of the pooled samples were run, each of them covering a 150 Th-wide fraction of the total 400–1000 m/z range using 6 m/z-wide overlapping isolation windows.

### 4.8. Proteomic Data Analysis

The quantitative analysis was performed in Encyclopedia 0.9 [41] using default settings. For the building of the GPF chromatogram library, a spectral library predicted by Prosit [42] from the UniProt rat reference proteome was used as the input. The quantitative samples were then analyzed using the empirically corrected GPF chromatogram library in Encyclopedia. The statistical evaluations were carried out using Perseus [43] software on text reports exported from Encyclopedia software. A *p* < 0.05 of the Welch’s *t*-test and 1.2< and <0.83-fold change values were used as the criteria for significant changes. Protein quantification data and annotations from the UniProt database can be found in Table 3.

### 4.9. Pathway Enrichment Analysis

Pathway enrichment analysis was performed with the web-based g:Profiler public server using the g:GOSt tool (https://biit.cs.ut.ee/gprofiler accessed on 10 January 2022) [44]. Functional enrichment profiling was carried out, defining *Rattus norvegicus* as the queried organism, using the g:SCS multiple testing correction method and *p* = 0.05 as a threshold value. For the visualization of the results, the output gem file of the over-represented Gene Ontology Biological Process and Cellular Component terms were loaded to Cytoscape v3.8.2 [45]. An enrichment map was created with the EnrichmentMap v3.3.2. application of Cytoscape based on the instructions of Reimand and colleagues with <0.1 adjusted *p*-value similarity score [46].

### 4.10. Functional Protein–Protein Interaction and Network Clustering Analysis

Protein–protein interaction analysis among the identified proteins was performed with the STRING v11.5 database (https://string-db.org ELIXIR, Cambridge, UK, accessed on 31 December 2021) [47] using the default 0.4 medium confidence and 5% false discovery rate (FDR) stringency values. Without any modification, the resulting protein–protein interaction chart was incorporated into Cytoscape and modified using the STRINGApp v1.7.0 and STRING Enrichment applications [48]. Cluster analysis was performed with clusterMaker2 v1.3.1 using the Markov Clustering Algorithm in Cytoscape. The clustering was based on the combined score calculated from the experimental and computational interaction values assigned by the STRING database.

### 4.11. Gene Set Enrichment Analysis of the Proteomic Dataset

Enriched gene sets from the whole, unfiltered proteomics data were explored with GSEA software v4.1.0 (joint project of UC San Diego, CA, USA and Broad Institute, Cambridge, MA, USA) [49]. After preprocessing to the requested format and the addition of appropriate phenotypes labels, the raw data were ranked according to the calculated fold change values, and GSEA was performed with all of the annotated ontology sets from the Molecular Signatures Database collections [50]. Only gene sets with FDR value 0.1 > were considered as significantly enriched. Interaction analysis was performed on only gene sets falling into FDR 0.1 > criteria and were visualized with Cytoscape as described in the previous sections. Subsequent leading-edge analysis was performed from the core enrichment proteins of the significantly enriched gene sets with the in-built plugin of the GSEA software.

### 4.12. KEGG Analysis and Visualization of Core Enrichment Proteins with Pathview

All the proteins that contributed to the core enrichment of the previously described GSEA analysis were further analyzed and classified according to the Kyoto Encyclopedia of Genes and Genome resource using Pathview Web (https://pathview.uncc.edu/home accessed on 17 January 2022) [51,52]. Proteins from the input list were colored according to the relative fold change values compared with the normocholesterolemic control group and then integrated into graphical images of the respective processes.

### 4.13. Statistics

All values are expressed as the mean + SEM (*n* = 6 in each group). Student’s *t*-test was used to evaluate the effect of the cholesterol-enriched diet on the plasma lipid levels and echocardiographic measurements. For proteomic data, a two-one-sided test was used for the equivalence test, and the statistical significance was tested using the unpaired Welch test. Multiple testing correction was applied in pathway enriched analysis. A *p* value < 0.05 was considered as an indicator of significant difference among the groups.

## 5. Conclusions

We conclude that hypercholesterolemia leads to alterations in the left ventricular proteome, affecting both the structural and metabolic protein networks. Cholesterol feeding seemed to modify the cardiac cytoskeleton arrangement and contractile apparatus, and also impaired mitochondrial function. These alterations in the proteome may provide a feasible explanation for the mild cardiac dysfunction observed in hypercholesterolemia. The results of our network and enrichment analyses might contribute to a better understanding of how alterations in the proteome may contribute to cardiac dysfunction in the presence of metabolic risk factors.

## Figures and Tables

**Figure 1 ijms-23-07387-f001:**
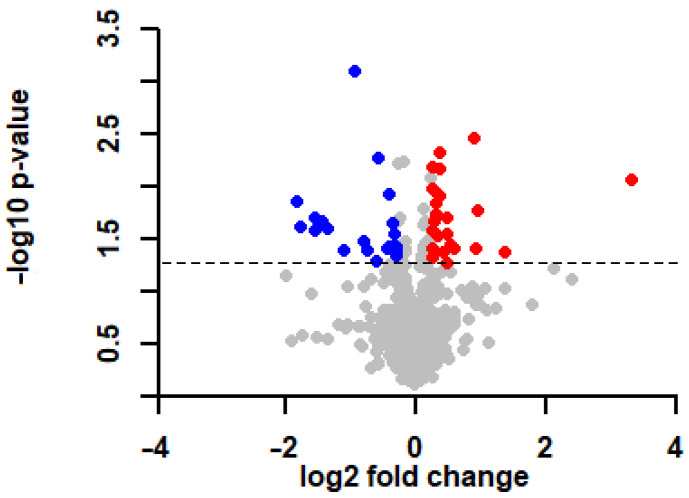
Diet-induced hypercholesterolemia leads to alterations in the myocardial proteome. Volcano plot showing the significantly (*p* < 0.05 of Welch’s *t*-test, 1.2< and <0.83-fold changes) altered proteins induced by hypercholesterolemia in the left ventricles of rats. Each dot represents one distinct protein. The downregulated (blue) and upregulated (red) proteins are highlighted in the plot. The dashed line indicates the threshold value of significance (−log_10_ 1.3<) The differentially expressed proteins are listed in Table 3.

**Figure 2 ijms-23-07387-f002:**
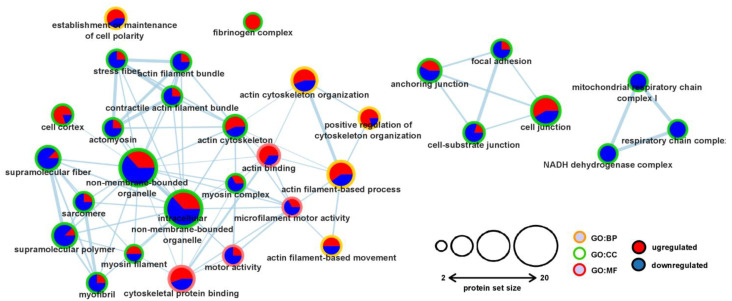
Pathway enrichment analysis revealed changes in the contractile and cytoskeletal systems. Significantly changed proteins were subjected to gene ontology (GO) and pathway enrichment analysis and visualized with Cytoscape v. 3.8.2. (Institute of Systems Biology, Seattle, WA, USA). The size of the nodes corresponds to the number of proteins falling into the respective category, while the GO terminologies are marked as differently colored borders (MF: molecular function, BP: biological process, CC: cellular component). Edges (lines) represent overlaps and functional interactions among the nodes. The width of each edge corresponds to the similarity score between the nodes. The numbers of up (red) and downregulated (blue) proteins were incorporated into the nodes as pie charts.

**Figure 3 ijms-23-07387-f003:**
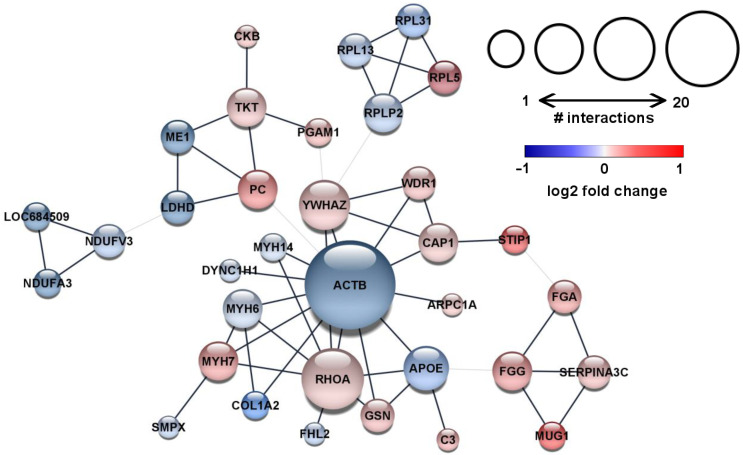
Downregulated beta-actin was established as a prominent hub of the protein interaction network in the hypercholesterolemic myocardium. Functional protein–protein interaction network and subsequent cluster analysis according to Markov Clustering Algorithm using the STRING database and the in-built plugin of Cytoscape. Each node (circle) represents one protein and is labeled according to gene IDs. The node size corresponds with the number of interactions of the respective protein. The color of each node indicates the fold change values (red for upregulated and blue for downregulated expression). Edges (grey lines) represent the interactions between nodes, with their thickness indicating interactions within the clusters (wide edges) or among the proteins in different clusters (narrow edges).

**Figure 4 ijms-23-07387-f004:**
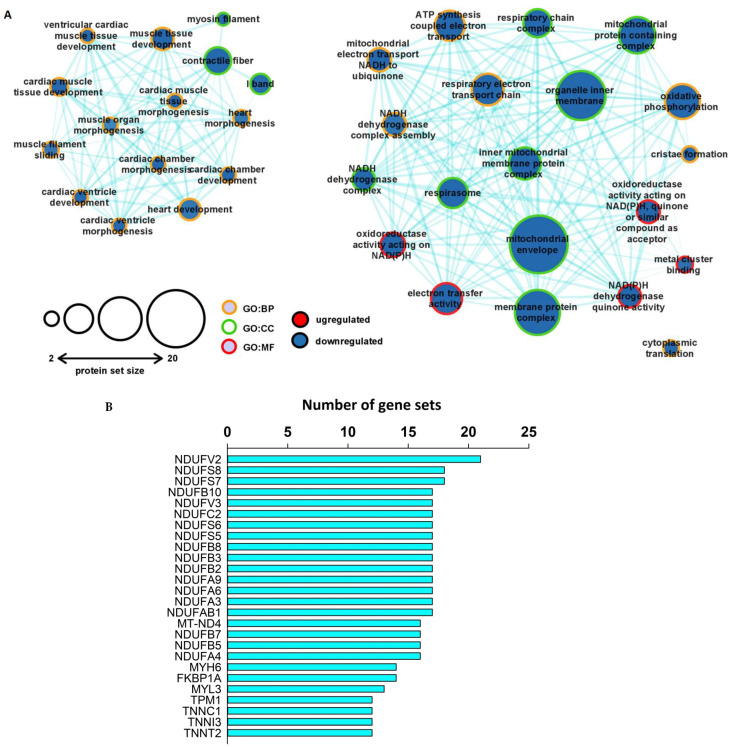
GSEA analysis of the unfiltered proteomic dataset revealed concordant downregulated expression patterns of proteins associated with the arrangement of the contractile and cytoskeletal system, as well as with the mitochondrial respiratory chain. (**A**) Enrichment map of the major GO sets influenced by hypercholesterolemia at the whole proteome level. Pathway enrichment analysis was performed with protein-specific GSEA. Circles represent enriched gene sets (nodes). The node size corresponds to the size of the number of proteins falling into a respective gene set category. The two node colors represent the trend of the quantitative change in the hypercholesterolemic left ventricle, while the color of the nodes’ border indicates the respective GO category. Edges represent similarity among the gene sets, as the thickness of each edge corresponds to the overlap between the nodes. The enriched gene sets were visualized with Cytoscape v3.8.2. (**B**) Output of the leading-edge analysis of the enriched GO terms at FDR < 0.1 performed with GSEA. The numbers on the horizontal axis indicate the number of appearances of the respective protein in the significantly enriched subsets. The graph was created with SigmaPlot v.12.0.

**Figure 5 ijms-23-07387-f005:**
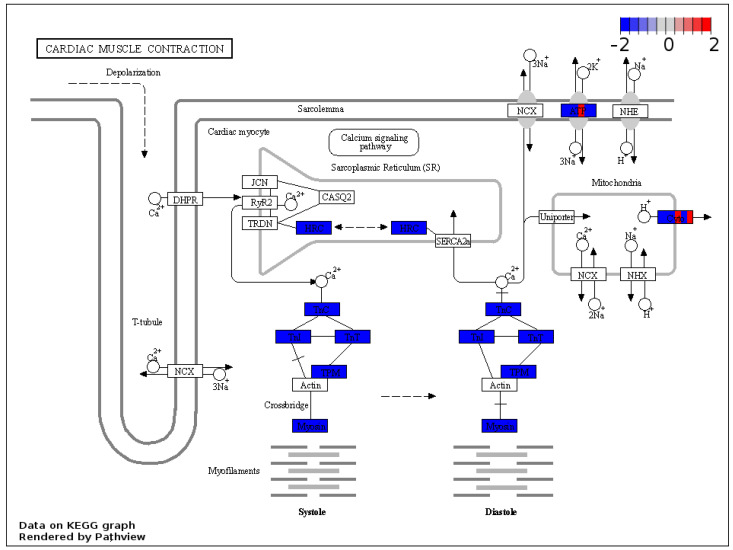
KEGG analysis showed concordant downregulated expression patterns of cardiac-specific troponins and the myosin complex. Visual representation of leading-edge protein subsets based on the Kyoto Encyclopedia of Genes and Genomes database. Each protein was divided into six sections and was colored based on the relative expression count compared with the normocholesterolemic group. Pathway graphs were created with Pathview Web (https://pathview.uncc.edu/home accessed on 17 January 2022).

**Figure 6 ijms-23-07387-f006:**
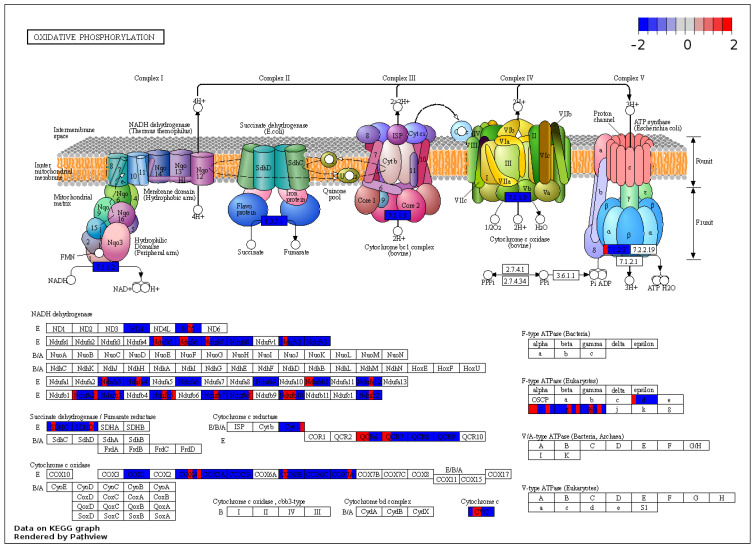
KEGG analysis showed concordant downregulated expression changes in protein components of the respiratory chain in the hypercholesterolemic left ventricle. Visual representation of leading-edge protein subsets based on the Kyoto Encyclopedia of Genes and Genomes database. Each protein was divided into six sections and was colored based on the relative expression count compared with the normocholesterolemic group. Pathway graphs were created with Pathview Web (https://pathview.uncc.edu/home accessed on 17 January 2022).

**Table 1 ijms-23-07387-t001:** Cholesterol-enriched-diet-induced high blood cholesterol. General characterization of normocholesterolemic (Normochol) and hypercholesterolemic (Hyperchol) rats after the 8-week diet period. * *p* < 0.05; *n* = 6.

	Normochol	Hyperchol
Body weight (g)	485 ± 22	521 ± 17
Tibia length (cm)	4.20 ± 0.08	4.21 ± 0.05
Left ventricular weight (mg)	1242 ± 42	1230 ± 48
Total cholesterol (mmol/L)	1.52 ± 0.11	4.35 ± 0.21 *
Total triglyceride (mmol/L)	0.44 ± 0.03	1.18 ± 0.08 *

**Table 2 ijms-23-07387-t002:** Diet-induced hypercholesterolemia leads to cardiac dysfunction. Left ventricular morphological and functional parameters examined by echocardiography after the 8-week diet period in both normocholesterolemic (Normochol) and hypercholesterolemic (Hyperchol) rats. Values are the mean ± SEM (*n* = 6), * *p* < 0.05. AWT: anterior wall thickness, d: diastolic, MV A: late (atrial) ventricular filling velocity measured at the mitral valve, MV E: early ventricular filling velocity measured at the mitral valve, e’: septal mitral annular velocity, EF: ejection fraction, FS: fractional shortening, IWT: inferior wall thickness, LVEDD: left ventricular end-diastolic diameter, LVESD: left ventricular end-systolic diameter, PWT: posterior wall thickness, s: systolic, and SWT: septal wall thickness.

	Normochol	Hyperchol	*p*-Value
AWTs (mm)	3.86 ± 0.01	3.74 ± 0.13	0.471
AWTd (mm)	2.03 ± 0.10	2.17 ± 0.13	0.420
IWTs (mm)	3.88 ± 0.12	3.62 ± 0.13	0.154
IWTd (mm)	2.39 ± 0.13	2.30 ± 0.14	0.660
PWTs (mm)	3.71 ± 0.09	3.70 ± 0.22	0.938
PWTd (mm)	2.46 ± 0.20	2.43 ± 0.13	0.902
SWTs (mm)	3.81 ± 0.05	3.69 ± 0.13	0.442
SWTd (mm)	2.18 ± 0.06	2.27 ± 0.13	0.548
LVESD (mm)	2.39 ± 0.10	2.85 ± 0.28	0.155
LVEDD (mm)	6.32 ± 0.30	6.72 ± 0.25	0.343
FS (%)	62.06 ± 1.43	63.89 ± 4.06	0.679
EF (%)	93.61 ± 0.63	90.56 ± 2.18	0.209
MV E velocity (m/s)	1.28 ± 0.19	0.80 ± 0.14	0.067
MV A velocity (m/s)	0.95 ± 0.16	0.50 ± 0.16	0.076
E/A	1.39 ± 0.06	1.81 ± 0.17 *	0.043
e’ (m/s)	0.06 ± 0.00	0.04 ± 0.00 *	0.005
E/e’	20.78 ± 3.38	20.08 ± 3.80	0.894
E deceleration time	79.00 ± 9.08	51.56 ± 6.12 *	0.031
Heart rate (1/min)	350.33 ± 10.89	323.50 ± 5.00 *	0.049

**Table 3 ijms-23-07387-t003:** List of hypercholesterolemia-induced significant alterations in the left ventricular proteins. Proteins with fold changes of >1.2 or <0.83 were considered as significant. The fold change values are shown as ratio pairs.

UniProt ID	Gene Symbol	Protein Name	Fold Change
P09895	*Rpl5*	60S ribosomal protein L5	2.60
Q03626	*Mug1*	Murinoglobulin-1	1.96
O35814	*Stip1*	Stress-induced-phosphoprotein 1	1.91
P09006	*Serpina3n*	Serine protease inhibitor A3N	1.90
P52873	*Pc*	Pyruvate carboxylase	1.54
P02680	*Fgg*	Fibrinogen gamma chain	1.45
P02564	*Myh7*	Myosin-7	1.42
P06399	*Fga*	Fibrinogen alpha chain	1.42
P01026	*C3*	Complement C3	1.35
D3ZWC6	*Sntb1*	Syntrophin, basic 1	1.31
P25113	*Pgam1*	Phosphoglycerate mutase 1	1.30
P29147	*Bdh1*	D-beta-hydroxybutyrate dehydrogenase	1.30
Q68FP1	*Gsn*	Gelsolin	1.29
P05545	*Serpina3k*	Serine protease inhibitor A3K	1.27
Q5RKI0	*Wdr1*	WD repeat-containing protein 1	1.27
P07335	*Ckb*	Creatine kinase B-type	1.25
A0A0G2K542	*Ugp2*	UTP--glucose-1-phosphate uridylyltransferase	1.22
Q99PD4	*Arpc1a*	Actin-related protein 2/3 complex subunit 1A	1.22
P50137	*Tkt*	Transketolase	1.22
D4A5W5	*Recql4*	RecQ-like helicase 4	1.22
P63102	*Ywhaz*	14-3-3 protein zeta/delta	1.21
P61589	*Rhoa*	Transforming protein RhoA	1.21
Q08163	*Cap1*	Adenylyl cyclase-associated protein 1	1.21
G3V885	*Myh6*	Myosin-6	0.83
Q925Q9	*Sh3kbp1*	SH3 domain-containing kinase-binding protein 1	0.83
F1LNF0	*Myh14*	Myosin heavy chain 14	0.83
F1M7L9		Uncharacterized protein	0.82
P38650	*Dync1h1*	Cytoplasmic dynein 1 heavy chain 1	0.81
Q925F0	*Smpx*	Small muscular protein	0.78
O35115	*Fhl2*	Four and a half LIM domains protein 2	0.77
P02401	*Rplp2*	60S acidic ribosomal protein P2	0.77
Q6PCU8	*Ndufv3*	NADH dehydrogenase [ubiquinone] flavoprotein 3	0.76
P41123	*Rpl13*	60S ribosomal protein L13	0.76
Q5XIG9	*Mtfp1*	Mitochondrial fission process 1	0.74
P02650	*Apoe*	Apolipoprotein E	0.68
P62902	*Rpl31*	60S ribosomal protein L31	0.66
C0KUC6	*Lims1*	LIM and senescent cell antigen-like-containing domain protein	0.61
Q924S5	*Lonp1*	Lon protease homolog	0.58
P02466	*Col1a2*	Collagen alpha-2(I) chain	0.52
P60711	*Actb*	Beta-actin	0.47
A0A0G2K1W9	*Ldhd*	Lactate dehydrogenase D	0.37
M0RB63	*LOC684509*	NADH-ubiquinone oxidoreductase B9 subunit	0.34
A0A0G2KAA3	*Ndufa3*	NADH:ubiquinone oxidoreductase subunit A3	0.34
P13697	*Me1*	NADP-dependent malic enzyme	0.34
Q9QZA6	*Cd151*	CD151 antigen	0.28

## Data Availability

All data reported in this paper will be shared by the lead contact upon request.

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
