# Peer review of "Diet-Induced Hypercholesterolemia Leads to Cardiac Dysfunction and Alterations in the Myocardial Proteome"

_ijms, 2022, doi:10.3390/ijms23137387_

Round 1

Reviewer 1 Report

The current manuscript is a well organized, well planned research work.

The abstract is well written.

The introduction needs to be written with more information, please mention clearly what exactly you are going to focus on, using what methods or models etc.

The figures need to improve.

Figure 5 is overly complex. It could be divided into 2 full page figures for better visualization and to make the text clearly visible.

Author Response

The current manuscript is a well organized, well planned research work.

The abstract is well written.

We are grateful to the reviewer for this positive opinion on our study.

The introduction needs to be written with more information, please mention clearly what exactly you are going to focus on, using what methods or models etc.

Upon the suggestion of the Reviewer we specified the focus and the research tool of the present study in the Introduction. Further details are provided in the Methods section.

The figures need to improve.

Figure 5 is overly complex. It could be divided into 2 full page figures for better visualization and to make the text clearly visible.

We changed the display and increased the size of Figures 2, 4 and 5 along with slight modifications in the legend and text on the images as an attempt to improve the Figures. Additionally, Figure 5 was separated into two different Figures (Figure 5 and Figure 6) thereby increasing their size as requested by the Reviewer. Nevertheless, final optimization of the Figures will be carried out by the production team, once our MS is accepted for publication. We also uploaded the images of each Figures in separate files.

Reviewer 2 Report

In this interesting study, the authors observed that cholesterol feeding leads to alterations in the left ventricular proteome and affects both structural and metabolic protein networks, by modifying the cardiac cytoskeleton arrangement and contractile apparatus and impairing mitochondrial function.

The manuscript is interesting and well written. It is methodologically correct. The conclusions are supported by the results. This reviewer raises a few issues that need to be addressed.

1- It is well known that a high-fat meal produces an increase in TNF-alpha associated with endothelial dysfunction (Nutr Metab Cardiovasc Dis. 2007 May;17(4):274-9. doi: 10.1016/j.numecd.2005.11.014.). This pro-inflammatory systemic condition and endothelial functional alteration may favor the cardiac dysfunction described by the authors. This issue and above reference should be added in discussion.

2- A section on the limitations of the study is missing at the end of the discussion.

Author Response

In this interesting study, the authors observed that cholesterol feeding leads to alterations in the left ventricular proteome and affects both structural and metabolic protein networks, by modifying the cardiac cytoskeleton arrangement and contractile apparatus and impairing mitochondrial function.

The manuscript is interesting and well written. It is methodologically correct. The conclusions are supported by the results. This reviewer raises a few issues that need to be addressed.

We are glad that the reviewer has a positive opinion on our MS and thinks our study is interesting and well written. 

1- It is well known that a high-fat meal produces an increase in TNF-alpha associated with endothelial dysfunction (Nutr Metab Cardiovasc Dis. 2007 May;17(4):274-9. doi: 10.1016/j.numecd.2005.11.014.). This pro-inflammatory systemic condition and endothelial functional alteration may favor the cardiac dysfunction described by the authors. This issue and above reference should be added in discussion.

We incorporated the suggestion of the Reviewer into the discussion part in page 14.

2- A section on the limitations of the study is missing at the end of the discussion.

We agree with the Reviewer that limitations of the study might be added to clarify the conclusions. Therefore, we have provided the limitations section of the present manuscript at the end of the discussion as requested by the Reviewer.
